# Transcriptomic Signatures and Molecular Pathways in Hidradenitis Suppurativa—A Narrative Review

**DOI:** 10.3390/ijms26167704

**Published:** 2025-08-09

**Authors:** Jasmine Spiteri, Dillon Mintoff, Laura Grech, Nikolai P. Pace

**Affiliations:** 1Centre for Molecular Medicine & Biobanking, University of Malta, MSD 2080 Msida, Malta; jasmine.spiteri.13@um.edu.mt (J.S.); laura.grech@um.edu.mt (L.G.); 2Department of Pathology, Faculty of Medicine & Surgery, University of Malta, MSD 2080 Msida, Malta; dillon.mintoff@gov.mt; 3Department of Dermatology, Mater Dei Hospital, MSD 2090 Msida, Malta; 4Department of Applied Biomedical Sciences, Faculty of Health Sciences, University of Malta, MSD 2080 Msida, Malta; 5Department of Anatomy, Faculty of Medicine & Surgery, University of Malta, MSD 2080 Msida, Malta

**Keywords:** hidradenitis suppurativa, transcriptome, pathomechanisms, targeted treatment

## Abstract

Hidradenitis suppurativa (HS) is a chronic, relapsing inflammatory dermatosis of the pilosebaceous unit characterized by nodules, abscesses, and dermal tunnels. Recent transcriptomic studies have implicated dysregulation of innate and adaptive immune responses, epidermal barrier dysfunction, and systemic metabolic alterations. This review synthesizes findings from 16 studies investigating the HS transcriptome using bulk and single-cell RNA sequencing. Differential gene expression analyses revealed extensive upregulation of inflammatory cytokines and chemokines, particularly in lesional and perilesional skin. These changes were also mirrored in non-lesional skin, suggesting diffuse immune dysregulation beyond visibly affected areas. Downregulated pathways include those involved in lipid metabolism, muscle contraction, and neuronal signaling, potentially linking HS to obesity, metabolic syndrome, and neuropsychiatric comorbidities. Single-cell transcriptomics confirmed the enrichment of keratinocytes and immune cells (B cells, plasma cells, M1 macrophages, and T cells) with proinflammatory profiles in HS lesions. Keratinocyte dysfunction further implicated a compromised epidermal barrier in disease pathogenesis. While transcriptomic studies have advanced mechanistic understanding and highlighted therapeutic targets—such as the IL-1β–TH17 axis and B cell signaling pathways—methodological heterogeneity limits cross-study comparisons. Integration of multi-omics data and standardized phenotyping will be essential to identify robust biomarkers, stratify HS subtypes, and guide personalized therapeutic approaches.

## 1. Introduction

Hidradenitis suppurativa is a chronic inflammatory dermatosis of the pilosebaceous unit, characterized by painful nodules, abscesses, and sinus tract formation. Lesions predominantly occur in intertriginous, apocrine gland-bearing skin such as the axillae, anogenital, and inframammary areas. Other symptoms include malodor, purulent discharge and scarring, and, in severe cases, skin mutilation [1,2,3,4]. Global prevalence estimates of HS vary extensively (0.0003–4.1%), with a pooled estimated prevalence of 0.4% and a higher disease burden in young adults, females, and African/African American individuals [5,6,7,8,9,10]. Contributory risk factors include lower socioeconomic status, tobacco use, obesity and comorbidities such as inflammatory bowel disease and autoimmune disorders [7,11,12].

Timely diagnosis and effective treatment of HS require a deeper understanding of its complex pathophysiology in relation to its broad clinical manifestations [13,14]. HS pathogenesis is characterized by dysregulated innate immunity and impaired epidermal differentiation, with inflammation, cutaneous dysbiosis, and hyperkeratinization representing core pathogenic features [4,15]. Targeting inflammatory pathways thus play a pivotal role, as shown by the promising clinical responses of biologic therapies that inhibit inflammatory cytokines, such as adalimumab (TNF-α inhibitor), secukinumab and bimekizumab (IL-17A/F inhibitors) [1,13].

Transcriptomic analysis enables the comprehensive characterization of dysregulated molecular pathways in HS, offering insights into pathogenic mechanisms, inflammatory signatures, and potential therapeutic targets [16]. The methodologies applied range from targeted techniques (qPCR) to biologically agnostic approaches such as microarrays, RNA sequencing (RNA-seq) and single-cell RNA sequencing (scRNA-seq). In the seminal study by Blok et al. (2016), gene expression profiling of lesional skin using microarrays revealed over 800 differentially expressed genes (DEGs) when compared to healthy controls, whereas no significant changes were observed in peripheral blood transcriptomes [17]. Notably, there was a negligible difference in the expression of components of the γ-secretase complex in lesional skin, despite the putative link between pathogenic variants in these loci and monogenic forms of HS [17,18,19,20]. Subsequent RNA-seq studies have identified broader and more subtle transcriptomic changes, involving gene networks related to immunity, antimicrobial response, and skin structure that extend beyond visibly involved skin to perilesional and non-lesional skin (i.e., clinically unaffected skin 2 cm and ≥10 cm away from an inflammatory lesion, respectively) [21,22,23]. ScRNA-seq also corroborated these findings, with HS samples demonstrating increased infiltration of immune cell subsets and expression of inflammatory cytokines in dermal tunnel keratinocytes [24,25,26].

Despite these advances, HS transcriptomic studies are constrained by small sample sizes, incomplete or inconsistent patient metadata, heterogeneity in demographic profiles and control matching, and variability in sample collection protocols and anatomical biopsy sites. Such factors can introduce bias and limit the generalizability of findings to specific HS endophenotypes [27]. Moreover, the influence of non-genetic variables—such as smoking, diet, microbiome variability, and hormonal status—further confounds cross-study comparisons and hampers the reproducibility and interpretability of transcriptomic signatures across cohorts.

Against this background, this review systematically collates gene expression studies related to HS to link differentially expressed loci to molecular mechanisms underlying disease pathogenesis. Specifically, we aim to (1) identify and evaluate published gene expression studies investigating DEGs in patients with HS; (2) compare DEGs across different tissue types (lesional, perilesional, and non-lesional skin, and whole blood), to determine overlapping and tissue-specific transcriptomic alterations; (3) identify significantly enriched biological pathways and molecular functions associated with upregulated and downregulated genes in HS; (4) evaluate scRNA-seq studies and assess cell-type-specific transcriptional changes; (5) discuss the potential clinical implications of identified molecular pathways.

## 2. Results

### 2.1. Study Characteristics and Data Extraction

The literature search identified 33 studies investigating the transcriptomic signature of HS. Following the exclusion of meta-analyses (n = 3) and studies reusing previously published datasets (n = 4) or not reporting fold-change values and *p*-values for DEGs (n = 10), 16 primary studies were selected (Appendix A) [17,21,22,24,25,28,29,30,31,32,33,34,35,36,37,38].

Sample sizes of patients with HS ranged from 3 to 34 per study (mean 18 ± 8 patients). RNA-seq validated by qPCR was the most frequently used method, with earlier studies using hybridization-based approaches or microarrays and more recent studies employing scRNA-seq (Figure 1A).

Demographic and related clinical data were variably reported, with ethnicity, family history of HS, comorbid disorders, and lifestyle factors often underreported. The HS cohorts predominantly consisted of adult females (mean age: 36 ± 4 years) with elevated body mass indices (>30 kg m^−2^), and a high proportion of smokers (Figure 1B). Caucasian and African American/African participants (Appendix A) constituted most of the patients across five studies. Although anatomical biopsy sites were unreported in nine studies, the most frequently analyzed tissue type was axillary skin (n = 5), followed by buttock and inguinal skin (Figure 1C), but no atypical biopsy sites were reported. Disease severity was mostly stratified using Hurley staging, with at least 60% of the patients from nine studies classified as moderate (Hurley II) severity.

The characteristics of healthy control groups, including sex and mean age, were only reported in seven studies. The control groups were primarily matched based on biopsy site, discounting other potential confounding factors such as age (mean age of controls: 55 ± 12 years).

### 2.2. Differentially Expressed Genes Across Tissues

#### 2.2.1. Overlapping and Unique DEGs Across Tissues

A comparative analysis of DEGs across HS lesional, perilesional, and non-lesional skin and whole blood identified both shared and tissue-specific gene expression changes relative to healthy control tissue (Figure 2). Lesional skin exhibited the most pronounced transcriptomic changes, followed by non-lesional skin which showed a significant number of unique downregulated genes. A total of 132 upregulated and 150 downregulated genes overlapped between blood and skin tissues.

Non-lesional skin displayed a distinct transcriptomic signature, with 235 downregulated genes that were simultaneously upregulated in lesional tissue, suggesting a dynamic regulatory shift in disease evolution. Conflicting regulatory patterns were observed for certain genes within the same tissue type, potentially indicating spatial or temporal heterogeneity. In peripheral blood, results were less consistent. Gudjonsson et al. (2020) reported 332 DEGs whilst Blok et al. (2016) detected none [17,28].

#### 2.2.2. Gene Categories and Immune Signatures

An analysis of gene categories using iDEP 2.0 revealed that protein-coding genes dominated the HS transcriptome (Figure 3A,B). Lesional skin was particularly enriched in immunoglobulin (IG) and T-cell receptor (TR) gene transcripts. In contrast, whole-blood samples showed a less pronounced IG gene signature. Pseudogenes and non-coding RNAs (lncRNA, miRNA, snoRNA, and snRNA) were significantly represented among DEGs, suggesting a potential role for post-transcriptional regulation in HS pathogenesis.

### 2.3. Over-Representation Analysis

#### 2.3.1. Upregulated Pathways

Pathway enrichment analysis of the upregulated DEGs using Reactome (Figure 4A) revealed a robust association with both innate and adaptive immune responses in HS skin. These pathways were predominantly driven by chemokine and interleukin (IL) signaling, namely IL-4 and IL-13 pathways, implicating a role for T helper (Th) cell-mediated immunity. The involvement of T_H_2 cytokine signaling is further underscored by the enrichment of pathways related to immunoregulatory interactions between lymphoid and non-lymphoid cells. Signaling of the pleiotropic cytokine IL-10 was also significantly enriched in both lesional and perilesional skin samples.

In non-lesional skin, there is a prominent dysregulation of chemokine binding and signaling, with significant enrichment of peptide ligand-binding receptor pathways and antimicrobial response pathways. Additionally, aberrant expression of cornified envelope components is conducive to disruption of epidermal barrier integrity. KEGG-based enrichment (Appendix A) largely corroborated the Reactome findings by confirming the upregulation of key inflammatory pathways, including IL-17 signaling and cytokine–cytokine receptor interactions across lesional and non-lesional tissues.

Differential gene expression analysis of whole-blood samples from HS cases revealed a distinct profile characterized by high false discovery rate (FDR) values and an absence of significant enrichment for canonical inflammatory pathways. Nonetheless, the DEGs identified in blood samples overlapped with genes involved in cellular stress responses, protein metabolism and translation regulation, and pathways associated with cell survival—particularly those involving tumor necrosis factor receptors (TNFRs). This pattern indicates a shift towards homeostatic and metabolic regulatory processes in the blood transcriptome of patients with HS, aimed at mitigating the dysregulation associated with chronic skin inflammation.

#### 2.3.2. Downregulated Pathways

In lesional skin, pathways downregulated according to Reactome prominently included muscle contraction, neuronal signaling, and lipid metabolism (Figure 4B). Neuronal signaling, particularly via SLIT and ROBO proteins, was also suppressed in non-lesional skin, where downregulation was more pronounced for pathways related to the electron transport chain and cellular respiration. These findings, which were also substantiated by the enriched KEGG pathways (Appendix A), suggest a role for mitochondrial dysfunction and impaired energy metabolism in HS pathogenesis.

In whole blood, downregulated pathways displayed high FDR values. Nevertheless, the downregulated genes were enriched in pathways related to immune response modulation, glycosaminoglycan metabolism and extracellular matrix composition, cellular turnover, and coagulation processes. These alterations point to a complex systemic phenotype in HS accompanied by a range of subtle systemic molecular perturbations.

### 2.4. Insights from Single-Cell Transcriptomics

Compared to bulk RNA-seq data derived from lesional skin, scRNA-seq has revealed distinct transcriptional signatures within keratinocyte populations—a pivotal cellular component in HS—characterized by lower representation of IG and TR genes (Figure 5A).

Nonetheless, pathway enrichment analysis from scRNA-seq data revealed consistent upregulation of immune-related processes within keratinocytes, particularly those associated with cytokine signaling cascades involving IL-10, type I interferons (IFN-α/β), and antimicrobial response genes (Figure 5B) [25]. Gudjonsson et al. (2020) further identified an inflammatory keratinocyte subset characterized by transcriptional activation of IFN-γ, IL-36γ, TNF, and IL-17A signaling axes [28]. Interestingly, the cornification pathway (although not statistically significant) was found to be simultaneously upregulated and downregulated within different keratinocyte subsets (Figure 5B), reflecting spatial and temporal heterogeneity in keratinocyte differentiation states. Moreover, concurrent downregulation of pathways related to keratinization and lipid metabolism reinforces the concept of epidermal barrier dysfunction.

Single-cell and bulk transcriptomic analyses consistently reveal the presence of B cells, T cells, plasma cells, monocytes/macrophages, mast cells, natural killer (NK) cells, and dendritic cells in inflamed HS tissue. The upregulation of genes involved in B cell recruitment (CXCL13, CXCR5) and survival (BAFF, APRIL), as well as their differentiation into plasma and memory B cells as disease progresses, further implicates humoral immunity in the HS inflammatory milieu [31,39]. Genes associated with proinflammatory M1 macrophages such as those involved in cytokine production and antigen presentation are markedly upregulated in HS lesions, while transcripts characteristic of the anti-inflammatory M2 phenotype are downregulated [25]. Together with the increased presence of type 1-polarized T cells in both lesional and non-lesional skin, as well as NK cells, this suggests a dominant type 1 inflammatory response that is largely orchestrated by type I and type II interferon (IFN) signaling pathways (IFN-α/β and IFN-γ) [25,31]. Additionally, monocytes/macrophages and dendritic cells exhibit heightened expression of antimicrobial and antiviral defense signatures [25].

## 3. Discussion

This review synthesizes findings from 16 transcriptomic studies investigating the molecular landscape of HS. The analysis reveals a complex and multifaceted HS pathophysiology, dominated by immune dysregulation, epidermal barrier dysfunction, and potentially metabolic alterations, while also highlighting significant research gaps.

### 3.1. Immune Dysregulation as a Central Feature

The findings from this review provide compelling evidence that immune dysregulation is a key driver of HS pathogenesis. The most consistent finding across studies is the significant upregulation of genes associated with innate and adaptive immune responses, particularly in lesional skin but also extending to perilesional and non-lesional skin, supporting the concept of a “pre-inflammatory state” in clinically unaffected skin. This is further substantiated by the enrichment of neutrophilic infiltration, psoriasiform hyperplasia, and activation of the IL-1β and IL-17 pathways in perilesional skin, despite the histological differences between the two tissues [16,33,40,41]. Activation of proinflammatory mediators involving the myeloid compartment, together with compromised immune regulation, have also been noted in non-lesional skin, suggesting that a reduced threshold for the initiation of myeloid cell-mediated inflammation plays a role in HS pathogenesis [31]. ScRNA-seq corroborated these findings, showing a substantial increase in immune cell populations with a predominant type I immune response that is also evident in keratinocytes of lesional and non-lesional skin, underscoring the role of innate immunity in HS pathogenesis [25,26,28,31,36].

The prominence of IL-10 signaling, while typically anti-inflammatory, may reflect a response to persistent inflammation (albeit insufficient), rather than indicating immune resolution [42]. Further supporting this hypothesis is the link between upregulated IL-10 and inhibition of IL-22, which ultimately results in the reduced expression of antimicrobial peptides, bacterial persistence in HS lesions and an attenuated immune response [43].

### 3.2. Epidermal Barrier Dysfunction

#### 3.2.1. Keratinocyte Involvement in Barrier Dysregulation

In addition to immune dysregulation, our review highlights dysregulated epidermal homeostasis in HS. This is evidenced by the simultaneous upregulation and downregulation of genes involved in keratinization (Figure 5B), which could reflect the changes in this process as lesions evolve, i.e., from follicular occlusion due to increased keratinocyte proliferation to the inflammatory state. Single-cell data suggest that keratinocytes within HS lesions are not homogeneous but span a spectrum of activation and differentiation states. Spatially distinct (dermal vs. epidermal) keratinocyte subpopulations differ in expression profiles [26]. Kurzen et al. demonstrated altered epithelial differentiation profiles in HS, and dysregulation of keratin expression has been suggested as a source of HS, draining sinus epithelium fragility and predisposing it to rupture and subcutaneous abscess formation [44,45]. It is hypothesized that cornification in HS is not uniformly impaired but dysregulated in a spatially compartmentalized manner. This intra-epidermal heterogeneity is further modulated by cytokine and microbiome-derived stimuli, which act to suppress barrier gene expression. This leads to a transcriptional mosaic that underlies barrier failure, hyperkeratosis, and chronic inflammation in HS, potentially explaining the observed keratinocyte intra-population variability.

Genome-wide association studies (GWASs) have identified HS risk variants with key roles in keratinocyte proliferation and differentiation. These include rs121908120-A in *WNT10A* and rs17103088-G in *TMED10*, which enhances γ-secretase complex activity and hyperkeratinization [46]. Additionally, the quantitative trait locus rs59532114-A in *BCL2* is associated with increased expression of the antiapoptotic regulatory protein BCL2 in the outer root sheath cells of hair follicles, which may induce/exacerbate inflammation, resulting in skin tissue damage [47].

Transcriptomic and proteomic analyses identified increased expression of components of the eIF4F translation initiation complex and its associated target oncogenic proteins, which have been implicated in follicular hyperproliferation and sinus tract formation, both hallmarks of advanced HS lesions [38]. Similarly, dysregulation of regulatory lncRNAs in patients with *NCSTN*-associated HS appears to influence keratinocyte proliferation, differentiation and migration, collagen biosynthesis, and oncogenesis [48]. Data from lipidomic studies also implicate sphingolipids and neutrophil-recruiting lipids in this pathomechanism through their role in hair follicle occlusion and inflammation [49,50,51].

At a cellular level, keratinocytes were noted to adopt a proinflammatory phenotype characterized by an increased production of CCL5, IP-10, and IL-1β cytokines. This initiates an inflammatory cascade, exacerbated by the altered pattern of AMPs, which further occludes the hair follicle infundibulum [52,53]. The upregulation of the cornified envelope precursor proline-rich protein-3 (SPRR3) in apocrine sweat gland and sebaceous gland ducts may also hinder the organization of lamellar lipids in the bilayer. In concert with changes in cytokeratins (KRT6/16/17) involved in inflammation progression and wound healing, these findings underscore the impairment of barrier function in patients with HS [4].

#### 3.2.2. Microbial Dysbiosis: A Contributor to Chronic Inflammation in HS

The skin microbiome in HS lesions is markedly altered, with a predominance of Gram-negative anaerobic bacteria such as *Prevotella*, *Porphyromonas*, and *Fusobacterium*, and a concurrent reduction in commensal species. This dysbiosis, particularly in biofilm-rich tunnels and sinus tracts, likely exacerbates chronic inflammation by sustaining IL-17 and TNF-α signaling [24,54,55,56,57]. Microbial metabolic activity, including altered tryptophan catabolism and decreased aryl hydrocarbon receptor (AHR) ligand production, may further disrupt immune homeostasis [58]. While causality remains unclear, the microbiome’s role as both a contributor to and amplifier of HS inflammation underscores the importance of host–microbe interactions in disease pathogenesis, as described in detail in recent review articles [52,53,59,60,61].

### 3.3. Comorbidities and Systemic Involvement

#### 3.3.1. Metabolic Alterations and Associated Comorbid Disorders

This review revealed some evidence of metabolic dysregulation, especially in the downregulated pathways in lesional skin. This suggests that HS is not solely a localized skin disorder but has broader systemic implications, potentially explaining the observed relationship between HS and its comorbidities.

One of the most frequent comorbidities is metabolic syndrome (MetS) [62,63,64,65,66]. It is strongly associated with HS, among other dermatoses, as evidenced by the dysregulated expression of proteins related to metabolism/obesity (PPARγ, IGF-1R, and irisin) [67]. Despite MetS typically arising in patients after HS onset, a recent two-sample bidirectional Mendelian randomization study suggested a unidirectional causal link wherein MetS increases HS risk, but not vice versa [62,63,64,68].

Central obesity is a key feature of MetS and is a strong risk factor for HS. The causal relation between excess adiposity and HS risk is supported by several studies [69,70,71,72,73,74,75,76]. The mechanistic link between obesity and HS is complex and multidimensional, but it includes (1) mechanical friction, enhancing follicular occlusion and rupture; (2) elevated skin fold temperature and humidity, promoting microbial growth; (3) meta-inflammation and a systemic proinflammatory state, characterized by altered adipokine signaling, with disrupted lipid and glucose homeostasis [69,73,74,77,78]. The resultant chronic subclinical inflammatory milieu in HS contribute to endothelial dysfunction and the formation of atherosclerotic plaques, thus enhancing the risk of developing cardiovascular disorders [40,79,80,81,82].

HS has also been linked to autoimmune disorders, such as multiple sclerosis, rheumatoid arthritis, and spondyloarthritis [83,84]. The pathogenic mechanism relating HS to these conditions is still unclear, but genetic predisposition, complement activation and a proinflammatory environment may be implicated [85,86,87]. Inflammation is also involved in the association between HS and fibromyalgia via amplified nociceptive signaling and central sensitization [88,89]. The higher prevalence of psychological comorbidities in patients with HS, namely anxiety and depression, may also be related to peripheral inflammation due to a reduced synaptic availability of monoamines through cytokines crossing the blood–brain barrier [90,91,92,93]. However, processes such as tryptophan catabolism via the kynurenine pathway and the NLRP3 inflammasome could also be involved [34,58,93].

#### 3.3.2. HS Blood Transcriptome

Despite these systemic manifestations, whole-blood transcriptomic data remain inconsistent. While early gene expression profiling by Hotz et al. (2016) and Blok et al. (2016) using microarrays found no significant differences in DEGs, later RNA-seq analyses by Gudjonsson et al. (2020) and Lowe et al. (2020) reported substantial dysregulation, especially in B cell-related genes [17,28,52,94,95]. Apart from the larger number of transcripts and splice isoforms that can be detected through RNA-seq, other methodological variations including disparities in patient demographics, stratification, medication, and lifestyle may represent latent confounders when interpreting blood transcriptomes [40,96,97,98,99].

The discrepancy may also reflect inherent limitations in using blood as a surrogate for assessing HS-specific cutaneous processes, since the dysregulation of biomarkers such as IL-6 was found to be less pronounced in blood compared to the skin [40]. This could, however, be the result of post-transcriptional regulation, and thus dysregulated transcripts identified in the skin may not be as pronounced in blood but may still influence the systemic proteomic profile [17]. Supporting this hypothesis is the correlation between the skin transcriptome/proteome and the blood proteome, particularly the dysregulation of proteins related to neutrophil-mediated inflammation, T_H_1/T_H_17 pathways, TNF, cytokine activity, and B-cell signaling [23,94,100,101,102,103]. This suggests active translation of mRNA into proteins that may diffuse into the blood, resulting in the systemic effects observed in patients with HS [23].

### 3.4. Comparison with Other Inflammatory Dermatoses and Implications for HS Treatment

Comparative transcriptomic analyses reveal both shared and unique molecular pathways among HS and the inflammatory skin conditions psoriasis and atopic dermatitis (AD). Shared pathways and genes involved in inflammation, including IFN, IL-1, IL-3, IL-6, IL-4/IL-13, IL-12, IL-17, IL-23, and the T_H_1/T_H_2/T_H_17 signatures, highlight common mechanistic underpinnings [22,28,31,33]. Notably, IL-36-induced neutrophil recruitment, and keratinocyte hyperproliferation in response to serpins and cytokeratins KRT6/16, as well as S100 antimicrobial and immune cell mediation activity, have been suggested as common pathogenic processes in these dermatoses [22,33,104,105].

Nonetheless, HS presents with a more proliferative signature, leading to keratinocyte hyperplasia [31,33]. This is evidenced by the following: (1) the significant enrichment of immune cells (B cells/plasma cells and macrophages) and their corresponding cytokines (CCL2, CXCL1, IL-6, IL-32, and IL-1β); (2) the increased upregulation of immunoglobulins and the innate immune pathways; (3) the lack of a dominant T_H_ axis typically observed in psoriasis/AD [22,28,31,33,40,105]. ScRNA-seq corroborated these findings, showing a predominant IL-1β–T_H_17 cell cytokine axis in HS compared to the IL-23–T_H_17 pathway in psoriasis [26]. The predilection for the T_H_17 axis may also be influenced by tobacco use, since nicotine modulates the cutaneous immune response by promoting the upregulation of proinflammatory cytokines and inducing T_H_17 cells (Figure 1B) [106]. Nonetheless, the direct role of tobacco use in HS pathophysiology is still unclear, with some studies reporting no association or differences in serum proteomic profiles [102,107].

These differences underscore the importance of understanding the unique pathomechanisms of HS to develop more targeted treatments. One could infer that patients with heightened inflammation because of B cell infiltration would respond better to BTK and SYK pathway inhibitors (acalabrutinib/ibrutinib, and fostamatinib) as opposed to TNF-α inhibitors [28,31]. Similarly, IL-1RA (anakinra) and anti-IL-1β antibodies (canakinumab), as well as IL-17 agents would be better suited for patients with HS with an IL1β-TH17 axis, rather than therapies aimed at lowering IL-12 or IL-23 expression (ustekinumab, and risankizumab and guselkumab, respectively) [26,40,108,109].

Unless treatments with a wide spectrum of application are approved, such as the recent use of a heat shock protein 90 inhibitor, identifying the HS subtype on an individual basis would better inform treatment and enhance its benefits [22,33]. The integration of multi-omics data with existing subtypes would significantly improve HS classification systems. Inter-rate subjectivity, lack of associated medical/molecular data, and exclusion of genes not related to the γ-secretase complex and inflammasome signaling, as well as gene-gene interactions, reduce the effectiveness of clinical stratification systems based on presented symptoms or genomic data [110,111]. Hence, more relevant and comprehensive subtypes and corresponding biomarkers can be determined through the inclusion of transcriptomic and proteomic data, facilitating diagnosis and treatment selection.

### 3.5. Methodological Considerations

A notable limitation of this review is the integration of findings across heterogeneous datasets rather than a formal, study-by-study quality assessment framework. While this approach allowed us to capture broader trends and insights across studies, it may have introduced variability stemming from differences in study design, methodologies, and cohort characteristics. This makes direct comparison challenging and affects the generalizability and statistical power of our findings. In addition, this review does not quantify effect sizes or assess the reproducibility of DEGs across datasets, especially when considering the inherent biases in RNA-seq methods resulting from batch effects, inadequate normalization methods, and multimapping due to the high proportion of pseudogenes. A limitation of our enrichment analysis is that over-representation analysis and multiple hypothesis correction were conducted independently for each tissue type without cross-tissue normalization. A meta-analytical approach analyzing raw data instead of published processed data might reduce these limitations and provide a larger harmonized dataset that could potentially implicate additional biological processes in HS pathophysiology.

Furthermore, the lack of standardized clinical data leads to difficulties in establishing precise correlations between molecular findings and clinical phenotypes, and hinders determination of the sequence of events and their impact on HS pathogenesis. The inconsistent reporting of clinical data, particularly lifestyle factors, disease severity/Hurley stage, sample biopsy site, and comorbidities, also poses significant challenges in assessing their impact on disease pathogenesis. Additionally, the infrequent use of well-matched controls and the lack of result validation by other methods also compromise the study’s validity and interpretation. The latter issue is further limited by the lack of proteomic and metabolomic data integration.

Although this review captured enrichment of immune, epidermal barrier, and metabolic pathways, several pathways with relevance to HS pathogenesis (Notch signaling, Wnt/β-catenin signaling, and *NCSTN*/γ-secretase complex function) were absent or underrepresented in the enrichment results. This likely reflects a combination of biological and technical limitations. Some key pathogenic drivers may be regulated at the level of protein activity, splicing, or in discrete cells not sufficiently captured in bulk RNA-seq datasets. Additionally, under-sampling of specific skin regions (e.g., follicular infundibulum, dermal tunnels) or early-stage lesions may have obscured or diluted early or cell-type-specific signals. The limited resolution of pathway annotations in public databases may contribute to reduced sensitivity for detecting less canonical or poorly annotated pathways. These limitations underscore the need for integrative multi-omics approaches and standardized, cell-resolved sampling to more fully characterize the HS transcriptome. Future research should focus on large-scale, well-designed studies with standardized protocols to address these limitations. Functional studies, ideally incorporating multi-omics analyses, are crucial to validate the observed associations between gene expression changes and HS pathogenesis and address key research gaps. These include the following: (1) the spatial and temporal variability in transcriptomes between patients and within the same patient, accounting for the local microenvironment, whose metagenome can fluctuate with metabolic and lifestyle changes; (2) annotation of dysregulated genes whose function is still unclear; (3) the impact of gene interactions and the role of non-coding transcripts and pseudogenes, possibly as epigenetic switches or post-transcriptional modulators; (4) the determination of HS subtypes and corresponding biomarkers to develop targeted therapeutic strategies.

## 4. Materials and Methods

The study selection process of this systematic review is summarized in Figure 6, which shows the Preferred Reporting Items for Systematic Reviews and Meta-Analyses (PRISMA) flow diagram.

### 4.1. Study Selection and Inclusion Criteria

An electronic literature search was conducted in Pubmed/MEDLINE and Google Scholar databases to identify the studies related to the HS transcriptome published between January 1990 to June 2024. The search was restricted to studies in the English language, using the following combinations of free text keywords and Medical Subject Headings terms together with the Boolean terms “AND” and “OR”: “hidradenitis suppurativa”, “acne inversa”, “Verneuil’s disease”, “transcriptome”, “DEGs”, “gene expression”, “pathogenesis/pathogenic mechanism”, “inflammation”, “RNA sequencing”, “single-cell sequencing”. Original studies investigating differentially expressed genes or proteins in varying tissues of patients with HS were included. Both case–control studies that used hybridization or high-throughput methods, and meta-analyses were considered. Studies that were not detected through the database search were selected via citation review and handsearching based on relevance.

### 4.2. Exclusion Criteria

Studies were excluded if they were (1) based on animal models, (2) carried out using only qPCR methodology targeting specific genes, (3) a re-analysis of data from previous studies, (4) summarized data presented without fold changes (FC) and *p*-values, and (5) conference proceedings, reviews, comments or editorial letters.

### 4.3. Data Extraction

The article titles and abstracts were screened and selected according to inclusion and exclusion criteria. Any discrepancies in the selection process were resolved by consensus of the investigators. The information retrieved from the selected articles included the following: (1) gene expression analysis method; (2) tissue studied; (3) patient and controls demographic data; (4) main outcomes of the study; (5) FC and *p*-values for reported DEGs with the cut-off values.

### 4.4. Data Analysis

Study and patient characteristics were summarized using plots/tables, with quantitative variables (e.g., age, BMI) presented as mean ± standard deviation and categorical variables (e.g., sex, smoking status) as percentages, derived from reported values.

DEG lists with their corresponding log_2_ fold change and *p*-values for each tissue type relative to healthy controls were compiled from the available datasets. The UpSet plot was generated using ComplexUpset (R v4.5.0) and the gene types for each tissue were determined using iDEP 2.0 [112]. Over-representation analysis was conducted independently for each tissue type compared to healthy controls using the WEB-based Gene SeT AnaLysis Toolkit (WebGestalt) with the Reactome functional pathway database (except for downregulated genes in perilesional skin, n = 6) [113]. The query parameters were set as follows: “gene/protein” analyte type; “gene symbol” ID type; “genome” reference set; “weighted set cover” redundancy removal; 5 minimum number of analytes for a category; 2000 maximum number of analytes for a category; “Benjamini–Hochberg” multiple test adjustment; “top 10” significance level based on FDR threshold; 10 categories expected from set cover. For each tissue-specific DEG list, the Benjamini–Hochberg procedure was applied separately to adjust for multiple testing within that group. This strategy controlled the false discovery rate (FDR) while respecting differences in gene list size and content across tissues. No cross-tissue normalization was applied, since the enrichment analyses were performed independently and plotted solely to illustrate the top significant pathways within each tissue type. The enriched pathways for each tissue type resulting from redundancy reduction according to the weighted set cover (except for downregulated DEGs in perilesional skin) were then directly plotted against the corresponding −log_10_ *p*-value. In addition to Reactome, pathway enrichment analysis was independently conducted using the KEGG_2021_Human library via Enrichr to ensure robustness and explore the cross-database consistency of the observed biological signals [114].

## 5. Conclusions

This review highlights the multifactorial nature of HS pathogenesis, driven by dysregulated immune responses, epidermal barrier function, and cellular metabolism. Transcriptomic data reveal widespread alteration involving keratinocytes, immune cells, and microbiome interactions that extends beyond the lesional skin, emphasizing the degree of perturbation caused by HS and hence the multitude of phenotypes in which it can present. Although RNA-seq studies and GWASs have provided key insight, inconsistencies between variant interpretation and transcriptomic/proteomic methodologies need to be addressed in order to identify causative links and potential biomarkers for the different patient strata and to develop targeted treatments. As precision dermatology advances, integrating multi-omics data into HS classification frameworks will be essential to guide treatment decisions and improve patient outcomes.

## Figures and Tables

**Figure 1 ijms-26-07704-f001:**
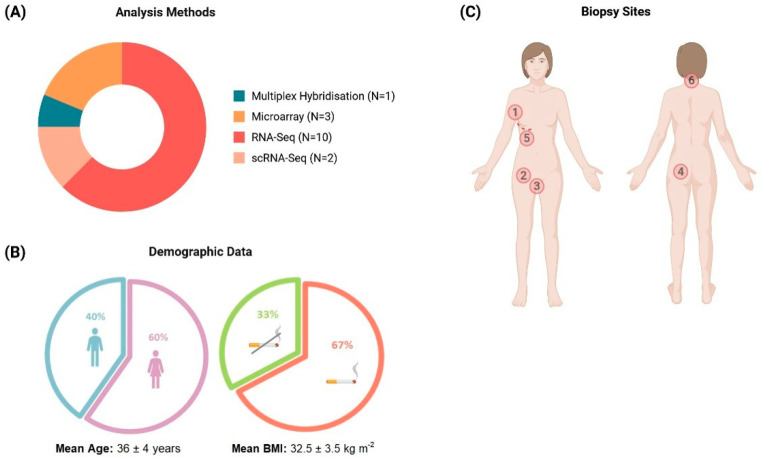
Overview of the methods and characteristics of patients in the reviewed studies. (**A**) Types and corresponding proportion of analytical methods used in the 16 studies. (**B**) Summary of the patients’ demographic data most reported among the studies, including sex, smoking status, age, and BMI (the mean values for age and BMI were calculated using the average values of the individual studies). (**C**) Areas from which HS skin samples were collected (the number also indicates the frequency of reporting of the anatomical site): 1—axilla (n = 6/16) [21,22,24,25,33,36], 2—inguinal (n = 5/16) [21,22,24,29,33], 3—pubic (n = 5/16) [21,22,24,29,33], 4—buttock (n = 3/16) [21,24,33], 5—inframammary (n = 3/16) [21,24,33], 6—neck (n = 2/16) [24,33].

**Figure 2 ijms-26-07704-f002:**
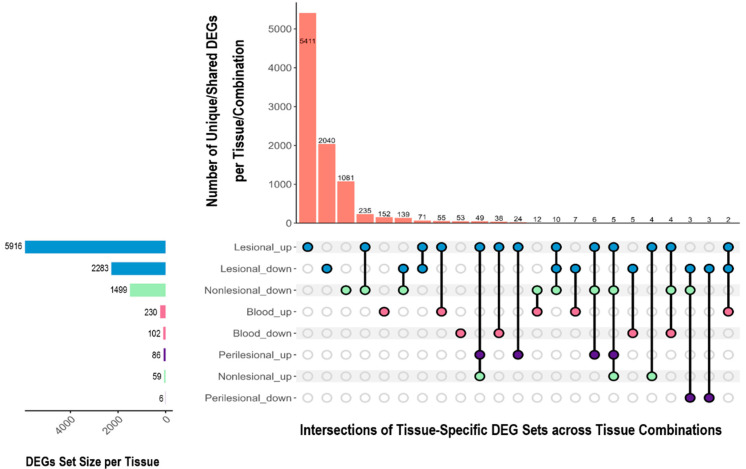
UpSet plot showing the overlap of differentially expressed genes (DEGs) across tissue types in hidradenitis suppurativa. The dot–line matrix below the vertical bars represents all possible intersections between tissue-specific DEG sets: each column represents a specific combination of tissue types (lesional skin—blue, perilesional skin—purple, non-lesional skin—green, and blood—pink), with filled dots indicating which tissues are included in that intersection. Vertical bars indicate the number of shared DEGs across those tissues. Horizontal bars depict the total number of DEGs identified within each individual tissue type, ordered by increasing DEG set size. The plot illustrates both unique and overlapping gene expression patterns, highlighting the extent of molecular dysregulation beyond clinically lesional skin.

**Figure 3 ijms-26-07704-f003:**
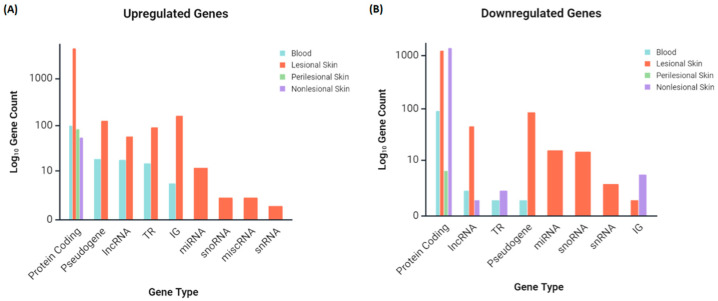
Exploratory assessment of the HS transcriptomes of skin and blood samples, showing the gene categories by tissue type for the (**A**) upregulated and (**B**) downregulated gene lists of patients with HS relative to healthy controls (IG—immunoglobulin gene, lncRNA—long non-coding RNA, miRNA—microRNA, miscRNA—miscellaneous RNA, snRNA—small nuclear RNA, snoRNA—small nucleolar RNA, TR—T cell receptor gene).

**Figure 4 ijms-26-07704-f004:**
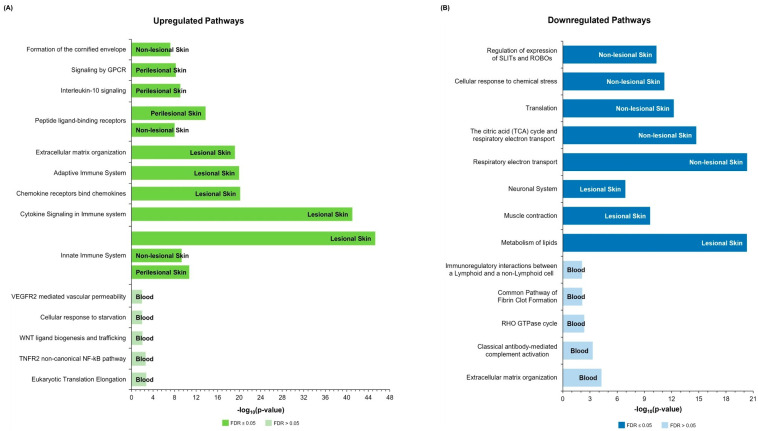
Enriched Reactome pathways in upregulated (**A**) and downregulated (**B**) DEG lists for each HS tissue type when compared to healthy controls; the pathways shown were obtained following weighted set cover redundancy reduction; all pathways had a *p*-value < 0.05, but those for blood tissue had a non-significant FDR value, i.e., >0.05.

**Figure 5 ijms-26-07704-f005:**
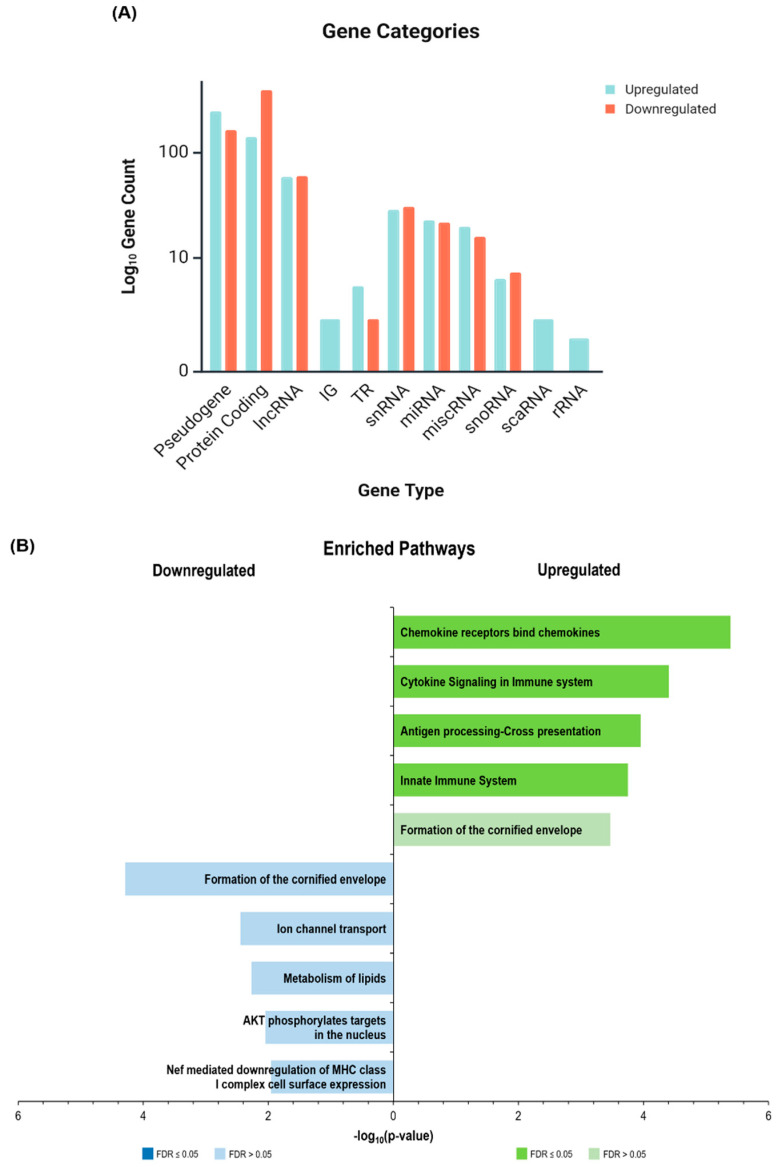
Single-cell transcriptome of keratinocytes. (**A**) Gene categories for upregulated and downregulated genes in the keratinocytes of patients with HS (IG—immunoglobulin gene, lncRNA—long non-coding RNA, miRNA—microRNA, miscRNA—miscellaneous RNA, rRNA—ribosomal RNA, scaRNA—small Cajal body-specific RNA, snRNA—small nuclear RNA, snoRNA—small nucleolar RNA, TR—T cell receptor gene). (**B**) Enriched Reactome pathways for upregulated and downregulated genes in the keratinocytes of patients with HS when compared to healthy controls. The pathways shown were obtained following weighted set cover redundancy reduction. All pathways had a *p*-value < 0.05, but all downregulated and 1 upregulated pathway had a non-significant FDR value, i.e., >0.05.

**Figure 6 ijms-26-07704-f006:**
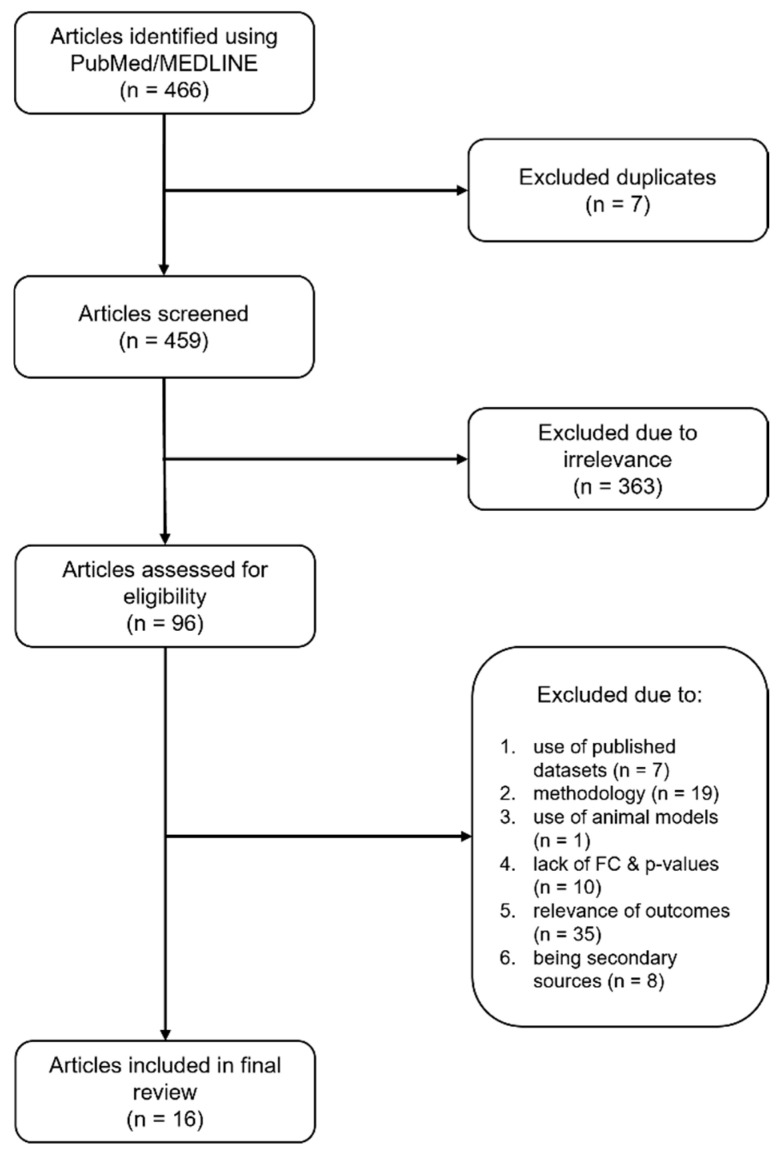
PRISMA flow diagram summarizing the article selection process.

## Data Availability

No new data were created or analyzed in this study. Data sharing is not applicable to this article.

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
