# Peer review of "Transcriptomic Signatures and Molecular Pathways in Hidradenitis Suppurativa—A Narrative Review"

_ijms, 2025, doi:10.3390/ijms26167704_

Round 1
Reviewer 1 Report
Comments and Suggestions for Authors
Dear authors
I had the pleasure to review your manuscript. My final opinion is that it would be a very good addition to the international literature about HS
Author Response
Thank you for reviewing our manuscript and for the positive comments.
Reviewer 2 Report
Comments and Suggestions for Authors
The authors carried out a systematic review of transcriptomic studies (bulk, single-cell RNA-sequencing) related to hidradenitis suppurativa (HS). Data from 16 publications have been compiled to identify differentially expressed genes in HS patients, compare their expression across lesional, perilesional, non-lesional skin and blood, identify enriched pathways and molecular functions, assess cell-type-specific changes from scRNA-seq, and explore their clinical implications. Authors have highlighted that transcriptomic analyses in HS revealed upregulation of inflammatory mediators across skin tissues, downregulation of metabolic and neuronal pathways, and enrichment of proinflammatory immune and keratinocyte populations, highlighting systemic immune dysregulation and epidermal barrier dysfunction.
2.1. Study Characteristics and Data Extraction
The characteristics of the study are indicated (type of analysis, sex, smoking status, age, BMI, anatomical site).
Comments:
- Figure 1 : Increasing the font size will make it easier for the reader to understand.
- Figure 1-C : To improve the interpretability of the anatomical site frequency representation, the number of studies corresponding to each site (eg: 1/16) could be indicated. The authors could add the associated references for each site.
- Lines 97-99: There is a numerical inconsistency in the description of the study selection process : the text indicates: 33 - 3 - 4 = 26, but only 16 studies are finally included. Did the author mean 23 core studies instead of 33 ? I couldn't find the answer in the PRISMA diagram, where these 2 numbers (33 or 23) are not present. More articles were excluded in this diagram (Material & Methods).
- While the introduction acknowledges the confounding role of smoking (line 80-83), the authors does not explore how smoking might specifically influence skin inflammation and transcriptomic profiles in HS although smoking status was reported as elevated in the cohort (67 %, Figure 1 B). It could be integrated to the Discussion to strengthen the interpretation of the findings.
2.2. Differentially Expressed Genes Across Tissues
Differentially expressed genes (DEGs) were extracted from published datasets. The authors then grouped the data into four groups : Lesional skin, Non-lesional skin, Perilesional skin, Whole blood.
An UpSet plot is used to visualize the overlap of DEGs between those different groups. This visualization is interesting because it enables us to identify both genes common to several groups (shared and potentially systemic molecular mechanisms) and genes specific to certain tissues (group-specific signatures).
Comments :
- The authors could explicitly reiterate in the Result section that the upregulated and downregulated statuses are defined relative to healthy tissue for each groups. Even if this is already stated in the Materials and Methods, a brief reminder in the main text would improve clarity and help readers interpret the findings more easily.
- Figure 2: Increasing the font size is necessary, as well as enriching the names of the axes:
- Horizontal axis (left): complete “Set Size” with Number of DEGs per tissue
- Vertical axis (bars): number of shared differentially expressed genes
- Horizontal axis (matrix): intersection of tissue-specific DEG sets
- Below the matrix: tissue combinations
- The author could also add a color code to distinguish tissue groups (blood, lesional skin, peri-lesional, non-lesional).
- The current legend could be enriched to guide the reader. For example :
- UpSet plot showing the overlap of differentially expressed genes (DEGs) across tissue types in hidradenitis suppurativa. The matrix of dots below the vertical bars represents all possible intersections between tissue-specific DEG sets: each column represents a specific combination of tissue types (lesional, perilesional, non-lesional skin, and blood), with filled dots indicating which tissues are included in that intersection. Vertical bars indicate the number of shared DEGs across those tissues. Horizontal bars depict the total number of DEGs identified within each individual tissue type. Tissues are ordered by decreasing DEG set size. […]
Gene categories and immune signatures
The authors perform a gene category analysis via iDEP 2.0.
Comments
- Figure 3: Two distinct sub-sections are brought together in this figure: 2.2.2. Gene Categories and Immune Signatures / 2.3. Over-Representation Analysis
- It is suggested splitting the figure in two or merging the subsections to improve clarity.
- Line 137 : although the use of iDEP 2.0 is mentioned in the “materials and methods” section, a brief recall in the body of the text would improve the text. For example: “An analysis of gene categories made with iDEP 2.0 [...]”.
2.3. Over-Representation Analysis
The authors perform an over-representation analysis using WEB-based Gene SeT AnaLysis Toolkit and the Reactome database.
Comments
- The enrichment analysis is limited to the Reactome database. It would have been interesting to compare the results with other databases (Enrichr, KEGG, or Gene Ontology).
- Figure 3C-3D: Please provide more clarity on the methodology that led to these figure panels. In particular:
- It is not specified whether the adjustment for multiple tests (Benjamini-Hochberg method) was applied within each tissue group individually or across all groups combined.
- Was any specific correction or normalization performed to homogenize the data across the four groups before representing them together on the same graph?
- Are there any biologically expected pathways missing from the findings? If so, this should have been addressed in the discussion (Methodological Considerations). It would have been valuable to evaluate the sensitivity of the review approach and to identify potential biological or technical limitations in detecting relevant gene expression signatures.
2.4 Insights from Single-cell Transcriptomics
- Figure 4 : The simultaneous up and downregulation of the cornification pathway in different keratinocyte subsets is an intriguing finding that reflects the heterogeneity within these populations (line 193). Although the discussion provides an overview of keratinocyte dysregulation (2.1), it does not explicitly address this intra-population variability. A more detailed explanation of this aspect or hypothesis could offer valuable insights into the dynamics of keratinocyte function in the disease.
Other issues of the review, such as small sample sizes and the absence of batch effect control, are appropriately discussed by the authors.
Author Response
The authors carried out a systematic review of transcriptomic studies (bulk, single-cell RNA-
sequencing) related to hidradenitis suppurativa (HS). Data from 16 publications have been
compiled to identify differentially expressed genes in HS patients, compare their expression
across lesional, perilesional, non-lesional skin and blood, identify enriched pathways and
molecular functions, assess cell-type-specific changes from scRNA-seq, and explore their
clinical implications. Authors have highlighted that transcriptomic analyses in HS revealed
upregulation of inflammatory mediators across skin tissues, downregulation of metabolic and
neuronal pathways, and enrichment of proinflammatory immune and keratinocyte
populations, highlighting systemic immune dysregulation and epidermal barrier dysfunction.
2.1. Study Characteristics and Data Extraction
The characteristics of the study are indicated (type of analysis, sex, smoking status, age, BMI,
anatomical site).
Comments:
Figure 1 : Increasing the font size will make it easier for the reader to understand.
Figure 1-C : To improve the interpretability of the anatomical site frequency representation,
the number of studies corresponding to each site (eg: 1/16) could be indicated. The authors
could add the associated references for each site.
Response: We have updated figure 1 to improve text clarity. The number of studies
corresponding to each anatomical site has been added to the text, along with the
respective references
Lines 97-99: There is a numerical inconsistency in the description of the study selection
process : the text indicates: 33 - 3 - 4 = 26, but only 16 studies are finally included. Did the
author mean 23 core studies instead of 33 ? I couldn't find the answer in the PRISMA
diagram, where these 2 numbers (33 or 23) are not present. More articles were excluded in
this diagram (Material & Methods).
Response: We have clarified this inconsistency in section 2.1 (Results). From 33 studies
investigating the transcriptomic signature of HS, we excluded metanalysis (n=3), studies
reusing previously published data (n=4) and 10 studies not reporting fold change and p-
values. As a result, 16 studies were integrated into our analysis.
While the introduction acknowledges the confounding role of smoking (line 80-83), the
authors does not explore how smoking might specifically influence skin inflammation and
transcriptomic profiles in HS although smoking status was reported as elevated in the cohort
(67 %, Figure 1 B). It could be integrated to the Discussion to strengthen the interpretation of
the findings.
Response: We have added a brief sentence on the role of smoking in modulating
cutaneous immune response in the discussion.
2.2. Differentially Expressed Genes Across Tissues
Differentially expressed genes (DEGs) were extracted from published datasets. The authors
then grouped the data into four groups : Lesional skin, Non-lesional skin, Perilesional skin,
Whole blood.
An UpSet plot is used to visualize the overlap of DEGs between those different groups. This
visualization is interesting because it enables us to identify both genes common to several
groups (shared and potentially systemic molecular mechanisms) and genes specific to certain
tissues (group-specific signatures).
Comments :
The authors could explicitly reiterate in the Result section that the upregulated and
downregulated statuses are defined relative to healthy tissue for each groups. Even if this is
already stated in the Materials and Methods, a brief reminder in the main text would improve
clarity and help readers interpret the findings more easily.
Response: We have reiterated in the results section 2.2.1 that upregulated and
downregulated statuses are defined relative to healthy tissue for each groups
.
Figure 2: Increasing the font size is necessary, as well as enriching the names of the axes:
Horizontal axis (left): complete “Set Size” with Number of DEGs per tissue
Vertical axis (bars): number of shared differentially expressed genes
Horizontal axis (matrix): intersection of tissue-specific DEG sets
Below the matrix: tissue combinations
The author could also add a color code to distinguish tissue groups (blood, lesional skin, peri-
lesional, non-lesional).
Response: Thank you for the suggestions, the UpSet plot in figure 2 has been updated to
improve clarity, add tissue-specific colour codes, axis labels.
The current legend could be enriched to guide the reader. For example :
UpSet plot showing the overlap of differentially expressed genes (DEGs) across tissue types
in hidradenitis suppurativa. The matrix of dots below the vertical bars represents all possible
intersections between tissue-specific DEG sets: each column represents a specific
combination of tissue types (lesional, perilesional, non-lesional skin, and blood), with filled
dots indicating which tissues are included in that intersection. Vertical bars indicate the
number of shared DEGs across those tissues. Horizontal bars depict the total number of
DEGs identified within each individual tissue type. Tissues are ordered by decreasing DEG
set size. […]
Response: The figure 2 legend has been updated accordingly.
Gene categories and immune signatures
The authors perform a gene category analysis via iDEP 2.0.
Comments
Figure 3: Two distinct sub-sections are brought together in this figure: 2.2.2. Gene Categories
and Immune Signatures / 2.3. Over-Representation Analysis
It is suggested splitting the figure in two or merging the subsections to improve clarity.
Response: Thank you for the suggestion, we have split the two sections into gene
category plots and enrichment plots to improve readability.
Line 137 : although the use of iDEP 2.0 is mentioned in the “materials and methods” section,
a brief recall in the body of the text would improve the text. For example: “An analysis of
gene categories made with iDEP 2.0 [...]”.
Response: This has been fixed.
2.3. Over-Representation Analysis
The authors perform an over-representation analysis using WEB-based Gene SeT AnaLysis
Toolkit and the Reactome database.
Comments
The enrichment analysis is limited to the Reactome database. It would have been interesting
to compare the results with other databases (Enrichr, KEGG, or Gene Ontology).
Response: We thank the reviewer for this valuable suggestion. To further validate our
findings, we conducted an additional enrichment analysis using the
KEGG_2021_Human library via Enrichr. The results are consistent with our Reactome-
based analysis and reinforce key immune pathways implicated in HS, including IL-17
and cytokine-cytokine receptor interactions. Notably, KEGG also highlighted disease-
associated signatures (e.g., prostate cancer, Parkinson disease) and metabolic
dysfunction, which corroborate some systemic features of HS. While Reactome provided
a more detailed view of cytokine sub-networks and immunoregulatory interactions,
KEGG complemented this by highlighting broader clinical relevance and shared
pathway architecture with inflammatory and metabolic diseases. We have now added
a) The following sentence to the methodology: ‘In addition to Reactome, pathway
enrichment analysis was independently conducted using the
KEGG_2021_Human library via Enrichr to ensure robustness and explore cross-
database consistency of the observed biological signals.’
b) The following sentence to the results section 2.3.1 ‘KEGG-based enrichment
corroborated the Reactome findings by confirming the upregulation of key
inflammatory pathways, including IL-17 signalling and cytokine–cytokine
receptor interactions, across lesional and non-lesional tissues.’
c) KEGG enrichment has been added as a separate supplementary file.
Figure 3C-3D: Please provide more clarity on the methodology that led to these figure panels.
In particular:
It is not specified whether the adjustment for multiple tests (Benjamini-Hochberg method)
was applied within each tissue group individually or across all groups combined.
Was any specific correction or normalization performed to homogenize the data across the
four groups before representing them together on the same graph?
Response: We thank the reviewer for this important methodological query. Over-
representation analysis was performed separately for each tissue type (lesional,
perilesional, non-lesional skin, and whole blood), and the Benjamini–Hochberg
correction for multiple testing was applied within each tissue-specific gene list
individually. This approach prevents inflation of false discovery rates due to differing
list sizes and avoids cross-tissue bias. No further normalization across tissue types was
applied prior to plotting, as the enrichment values are derived from independent gene
sets and are not directly compared in a statistical sense. Instead, the graphs are intended
to visually summarize the most significantly enriched pathways within each tissue,
based on -log10 p-values post Benjamini Hochberg correction. We have clarified this
steps in the revised Methods section.
In addition, in the methodological consideration section, we have added the following
limitation ‘A limitation of our enrichment analysis is that over-representation analysis
and multiple hypothesis correction were conducted independently for each tissue type
without cross-tissue normalization. A meta-analytical approach analyzing raw data
instead of published processed data might reduce these limitations and provide a larger
harmo-nized dataset that could potentially implicate additional biological processes in
HS pathophysiology.’
Are there any biologically expected pathways missing from the findings? If so, this should
have been addressed in the discussion (Methodological Considerations). It would have been
valuable to evaluate the sensitivity of the review approach and to identify potential biological
or technical limitations in detecting relevant gene expression signatures.
Response: Thank you for the insightful comment. We acknowledge that several
biologically plausible pathways, such as Notch and Wnt/β-catenin signalling which are
supported by genetic studies of HS were underrepresented in the pathway enrichment
results. This discrepancy may reflect both biological and technical factors. First, the
expression of certain disease-relevant genes (e.g. γ-secretase components) may be
spatially or temporally restricted, or regulated at the post-transcriptional or post-
translational level, making them less likely to be captured in bulk or scRNA-seq data.
Second, heterogeneity in sampling sites, cell-type composition, small cohort sizes, and
limited inclusion of early/pre-lesional skin may have reduced sensitivity for detecting
such signatures. Finally, pathway databases differ in annotation depth for under-
characterized pathways, which may also contribute to these omissions.
We have now expanded the Methodological Considerations section to explicitly discuss
the sensitivity and scope of our review approach, and the potential implications of
missing key pathways
2.4 Insights from Single-cell Transcriptomics
Figure 4 : The simultaneous up and downregulation of the cornification pathway in different
keratinocyte subsets is an intriguing finding that reflects the heterogeneity within these
populations (line 193). Although the discussion provides an overview of keratinocyte
dysregulation (2.1), it does not explicitly address this intra-population variability. A more
detailed explanation of this aspect or hypothesis could offer valuable insights into the
dynamics of keratinocyte function in the disease.
Response: Thank you for this observation which merits deeper discussion. We have now
expanded the Discussion to incorporate mechanistic insights from Johnston et al. (2021),
who propose that HS is characterized by a collapse of the terminal differentiation
program in the follicular infundibulum, leading to structural disorganization and
impaired cornification.
Other issues of the review, such as small sample sizes and the absence of batch effect control,
are appropriately discussed by the authors.
Reviewer 3 Report
Comments and Suggestions for Authors
The manuscript entitled “Transcriptomic Signatures and Molecular Pathways in Hidradenitis Suppurativa – A Narrative Review” presents a well-conducted and timely synthesis of the current transcriptomic evidence on hidradenitis suppurativa (HS), offering a multidimensional perspective on its pathogenesis. The review is clearly written and contributes meaningfully to the literature on this complex inflammatory dermatosis. The authors reported findings from 16 transcriptomic studies, including both bulk and single-cell RNA sequencing approaches, in order to map the molecular alterations underlying HS. This synthesis is not only comprehensive but also critically framed, offering thoughtful interpretations of the findings and situating them within broader discussions on immune dysregulation, epidermal barrier dysfunction, metabolic alterations, and systemic involvement. This review bridges basic molecular findings with potential clinical implications. The discussion linking gene expression patterns to inflammatory pathways, particularly the IL-1β–Th17 axis, B cell signaling, and interferon-related responses, is highly relevant in the current therapeutic landscape. The manuscript effectively suggests how transcriptomic insights could contribute to patient stratification and future personalized treatment strategies. This translational orientation significantly enhances the impact of the work. Moreover, the integration of single-cell RNA-seq data adds an important layer of resolution, revealing cell-type-specific alterations, particularly in keratinocytes, macrophages, and lymphocytes. The methodology of the review is appropriate and transparent. Another notable strength is the careful attention given to methodological heterogeneity across studies. The discussion is particularly rich and well-argued. It not only highlights known mechanisms, such as the role of cytokines, keratinocyte dysregulation, and microbial dysbiosis, but also addresses emerging areas of research. In conclusion, this is a rigorous and impactful review that successfully integrates complex molecular data into a coherent narrative of HS pathogenesis. It is relevant both to researchers in molecular dermatology and to clinicians seeking to understand the biological rationale for emerging therapies. The authors demonstrate an impressive command of the subject matter and present their findings with scientific maturity and clinical awareness. It is suitable for publication in its current form.
Author Response
The manuscript entitled “Transcriptomic Signatures and Molecular Pathways in Hidradenitis
Suppurativa – A Narrative Review” presents a well-conducted and timely synthesis of the
current transcriptomic evidence on hidradenitis suppurativa (HS), offering a
multidimensional perspective on its pathogenesis. The review is clearly written and
contributes meaningfully to the literature on this complex inflammatory dermatosis. The
authors reported findings from 16 transcriptomic studies, including both bulk and single-cell
RNA sequencing approaches, in order to map the molecular alterations underlying HS. This
synthesis is not only comprehensive but also critically framed, offering thoughtful
interpretations of the findings and situating them within broader discussions on immune
dysregulation, epidermal barrier dysfunction, metabolic alterations, and systemic
involvement. This review bridges basic molecular findings with potential clinical
implications. The discussion linking gene expression patterns to inflammatory pathways,
particularly the IL-1β–Th17 axis, B cell signaling, and interferon-related responses, is highly
relevant in the current therapeutic landscape. The manuscript effectively suggests how
transcriptomic insights could contribute to patient stratification and future personalized
treatment strategies. This translational orientation significantly enhances the impact of the
work. Moreover, the integration of single-cell RNA-seq data adds an important layer of
resolution, revealing cell-type-specific alterations, particularly in keratinocytes, macrophages,
and lymphocytes. The methodology of the review is appropriate and transparent. Another
notable strength is the careful attention given to methodological heterogeneity across studies.
The discussion is particularly rich and well-argued. It not only highlights known mechanisms,
such as the role of cytokines, keratinocyte dysregulation, and microbial dysbiosis, but also
addresses emerging areas of research. In conclusion, this is a rigorous and impactful review
that successfully integrates complex molecular data into a coherent narrative of HS
pathogenesis. It is relevant both to researchers in molecular dermatology and to clinicians
seeking to understand the biological rationale for emerging therapies. The authors
demonstrate an impressive command of the subject matter and present their findings with
scientific maturity and clinical awareness. It is suitable for publication in its current form.
Response: Thank you for reviewing our manuscript and for the positive comments.